# HIERARCHY-AIDED SPARSE ATTENTION FOR FAST LLMS PREFILLING INFERENCE

## ABSTRACT

Pre-filling Large Language Models (LLMs) with long-context inputs is computationally expensive due to the quadratic complexity of full attention. While global attention is essential during decoding, its importance diminishes during pre-filling, where the focus is on contextualizing tokens rather than predicting the next one. Building on prior work, we apply diagonal block sparse attention during the pre-filling phase, reducing attention-related FLOPs by over 90% without significant degradation in language modeling performance. To address the remaining performance gap, we propose **H**ierarchy-**A**ided **S**parse **A**ttention (HASA), which incorporates a specialized transformer branch. This branch extracts global embeddings from each chunk and aligns local attention with full-attention, facilitating cross-chunk interaction. HASA stabilizes sparse attention computations, making the pre-filling phase highly efficient, particularly in long-sequence scenarios. While HASA significantly accelerates the pre-filling phase, we ensure robust language modeling performance by enabling interaction between global embeddings across chunks, which prevents the performance degradation typically observed in sparse attention mechanisms. Given that there are limited methods specifically accelerating pre-filling, our baselines include various open-source long-context models. Across multiple benchmarks, HASA not only maintains performance but also outperforms baseline models in certain scenarios. We will release the models upon acceptance.

## 1 INTRODUCTION

In recent years, there has been rapid progress in long-context Large Language Models (LLMs), with performance now on par with commercial deployment standards. However, the conventional causal attention mechanisms continue to drive up inference costs, impeding their broader adoption. Causal attention's main challenges are the sharp increases in maximum GPU memory allocation and end-to-end latency, both of which scale quadratically with the sequence length. In reality, the memory allocation issue has been effectively tackled by researchers such as Rabe & Staats (2021); Dao (2024), who have shown that the

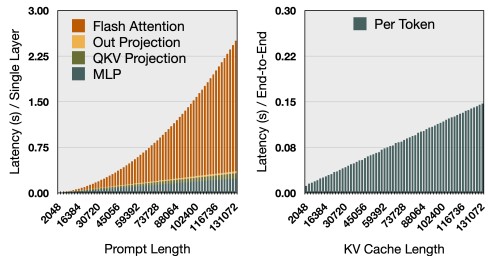

(a) Pre-filling phase.  (b) Decoding phase.
Figure 1: Inference latency of LLaMA2-7B on a machine with 8 RTX 3090 GPUs.

attention module can avoid quadratic memory scaling. Leveraging these insights, memory-efficient CUDA kernels like Flash Attention 2 (Dao, 2024) and PyTorch's SDPA (Guessous, 2024) have been introduced, effectively alleviating memory bottlenecks. However, the challenge of quadratic latency scaling remains unresolved, with no widely accepted solution currently on the horizon.

Given the unresolved challenge of quadratic latency scaling, we conduct an in-depth analysis to investigate its potential impact and the practical value of addressing this issue. Our analysis takes a user-centric perspective. From the user's standpoint, latency comprises two main components. The first is time to first token (TTFT), which indicates the pre-filling latency—the time needed to process the prompt and produce the initial token. The second component is time per output token (TPOT), representing the latency during the decoding phase—the time required to generate

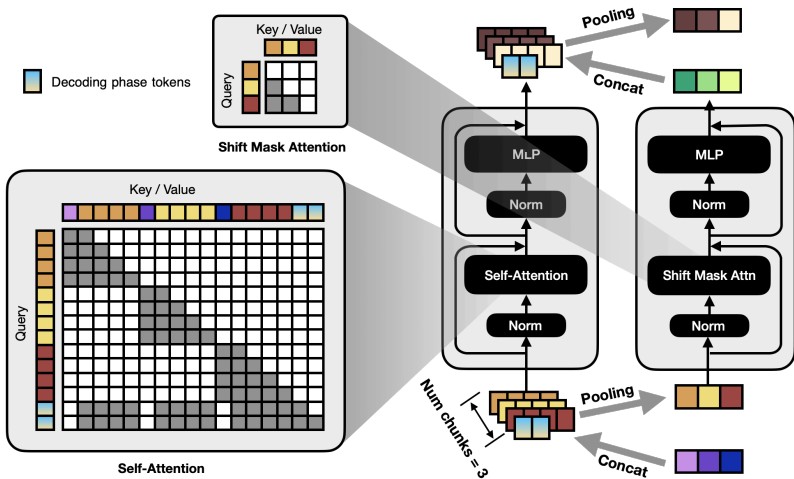

Figure 2: Overview of our proposed hierarchical transformer architecture.

each token sequentially in an auto-regressive process. Based on these components, the total latency experienced by the user can be expressed as the sum:

$$\text{Latency} = \text{TTFT} + \mathcal{C} * \text{TPOT}, \tag{1}$$

where $\mathcal{C}$ represents the number of tokens generated by the chat assistant, assumed to be constant.

To better understand the impact of sequence length on latency, we analyze the scaling behavior of TTFT and TPOT individually. As shown in Figure 1, TTFT exhibits quadratic growth with sequence length, reaching up to 80 seconds for prompts with 128K tokens. This suggests that users may experience delays exceeding one minute before receiving the initial token, potentially degrading their interaction experience. Conversely, TPOT scales linearly with sequence length, reaching approximately 0.15 seconds per token, even with a key-value (KV) cache of 128K tokens. Thus, TTFT is the sole component in the latency equation that exhibits quadratic scaling. As sequence lengths increase, this quadratic growth in TTFT poses a significant challenge to the deployment of long-context LLMs, leading to substantial delays in generating the first token, which must be addressed for scalability. Therefore, our work addresses the question: *How can we accelerate the pre-filling phase to mitigate TTFT's quadratic scaling without compromising model performance?*

Building on this inquiry, we revisit pioneering research on efficient pre-filling (de Jong et al., 2023; Ivgi et al., 2023), which suggests that while global attention is crucial for decoding, it is less important during pre-filling. In this phase, the goal is to convert each token into its contextual representation rather than predict the next token. This makes global attention less critical, especially in long-context tasks like retrieval-augmented generation (RAG), where retrieved passages are largely independent. Therefore, applying local attention during the pre-filling phase is highly reasonable. Inspired by this, we conducted preliminary experiments on LLaMA2-7B (Touvron et al., 2023a), applying diagonal block sparse attention during the pre-filling phase. We found that this method did not result in significant degradation of language modeling performance, while reducing attention-related FLOPs by over 90%. To further close the performance gap between this naive method and full-attention pre-filling, we re-examined earlier hierarchical attention methods (Guo et al., 2022; Zhu & Soricut, 2021). Building on these ideas, we introduce **H**ierarchy-**A**ided **S**parse **A**ttention (HASA), which incorporates a specialized branch to recover the global information lost in diagonal block sparse attention. Specifically, we extract a global embedding from each chunk and process the sequence of global embeddings through this specialized branch, encouraging interaction and fusion among them. These global embeddings are then prepended to their respective chunks and used to modulate attention scores. By calibrating the attention scores for each chunk, we align local attention with full-attention, stabilizing sparse attention computation and enabling the model to maintain strong language modeling performance even when processing long sequences.

We implement HASA on LLaMA2-7B (Touvron et al., 2023a) and LLaMA2-7B-32K (Together.AI, 2023a), fine-tuning them using the LoRA method (Hu et al., 2022). Our approach is evaluated across

diverse benchmarks, encompassing language modeling, few-shot natural language understanding (NLU) tasks, the Needle in a Haystack test (gkamradt, 2023), and LongBench (Bai et al., 2024). The experimental results indicates that our method not only expedites pre-filling and substantially decreases TTFT but also maintains, and in some cases, enhances model performance.

## 2 RELATED WORKS

**Causal Attention.** Causual attention is a parameterized module that takes in three inputs, $Q, K, V$, each of size $N \times D$, where $N$ is the sequence length and $D$ is the model's dimension. It works as:

$$\text{CausalAttention}(Q, K, V) = \big(\text{softmax}(QW_qW_k^\top K^\top) \odot M\big) VW_v, \tag{2}$$

where $M$ is a matrix that only allows information to flow from earlier to later parts of the sequence (like a one-way street). We skip some details for now, such as normalization after softmax, but in the actual code, we do some extra steps to make sure everything works smoothly. In a typical setup, if $X$ is the input data, the output is calculated as $\text{CausalAttention}(X, X, X)$. This process has a complexity of $\mathcal{O}(N^2D)$, which means it gets more demanding as the sequence gets longer.

There are special tools like Flash Attention (Dao, 2024) and PyTorch SDPA (Guessous, 2024) that use powerful computing techniques to speed things up. These have helped a lot, but they don't change the basic problem that the complexity grows with the square of the sequence length.

**Efficient LLMs.** Before the advent of LLMs, there were efficient transformer models that used sparse or linear attention (Wang et al., 2020; Zaheer et al., 2020) But now, training LLMs is very expense, so it's smarter to improve existing LLMs rather than building new ones from the start. LM-Infinite (Han et al., 2024) and Streaming-LLM (Xiao et al., 2024) use a lambda-shaped attention that deals with long texts by keeping the time and memory needed constant. But they might skip over important words. Techniques such as $H^2O$ (Zhang et al., 2023b), Scissorhands (Liu et al., 2023), and FastGen (Ge et al., 2024) make LLMs faster by reducing the size of KV cache. They do this by making the attention less dense. Other innovations, such as Multi-Query Attention (Shazeer, 2019) and Grouped-Query Attention (Ainslie et al., 2023), use fewer attention heads to save memory. Methods like OmniQuant (Wenqi et al., 2024) and AffineQuant (Ma et al., 2024) make LLMs faster by the quantization. While these methods speed up LLMs, they only reduce the complexity in certain parts, not the overall problem of handling long sequence, which remains challenging.

## 3 PRUNING ATTENTION FOR EFFICIENT PRE-FILLING

**Is Full Attention Computation Essential for Pre-filling Tokens?** In early sparse attention models like Longformer (Beltagy et al., 2020) and Big Bird (Zaheer et al., 2020), inputs are categorized into global tokens and local tokens. Global tokens leverage full attention to access the entire sequence, crucial for downstream tasks, while local tokens apply local attention to generate context from their surroundings. This methodology has proven effective for long-sequence modeling (Joshi et al., 2017; Yang et al., 2018; Tu et al., 2019), maintaining model capacity without inefficient attention. Drawing inspiration from this, we question whether similar concepts could be applied to LLMs. For example, during the pre-filling phase, token embeddings serve only as context for later decoding, not for token generation. Might we then use sparse local attention for these tokens exclusively for pre-filling tokens?

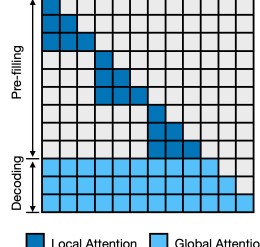

Figure 3: Diagonal block sparse attention with a chunk size $S = 3$.

After pre-filling, we could revert to full attention during generation phase, as these tokens will predict subsequent tokens. The success of sparse attention as a viable alternative to full attention supports our preliminary investigation into applying it within LLMs.

One of the main challenges resulting from the sparse attention is that many implementations (Jiang et al., 2024; Yao et al., 2024) require specialized kernel optimizations for efficient computation. To simplify the deployment, we propose computing only the causal attention weights within the diagonal block of attention matrices, as depicted in Figure 3. This method allows for straightforward implementation by segmenting the pre-filling sequence to multiple chunks of equal size $S$ and applying casual attention only among tokens within each chunk.

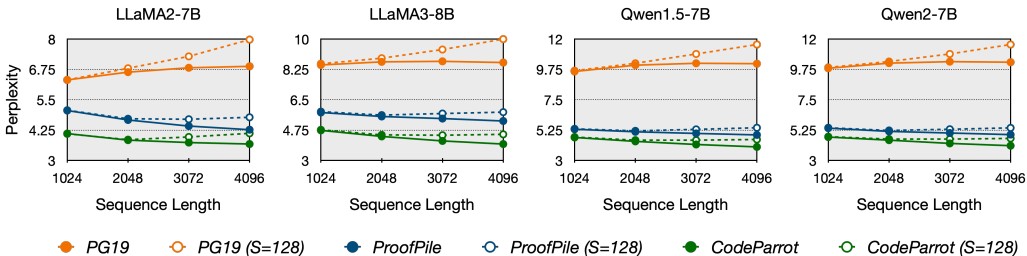

Figure 4: Language modeling performance of different models using a chunk size $S = 128$.

**Chunk Pre-filling.** This chunking strategy is a commonly utilized technique for accelerating pre-filling processes. The Fusion-in-Decoder (FiD) (Izacard & Grave, 2021) is pioneering in leveraging chunk pre-filling to expedite the pre-filling process, primarily for open-domain question answering where unrelated articles could be naturally segmented into chunks of varying lengths for individual encoding. SLED (Ivgi et al., 2023) expands the applicability of FiD by dividing the text into uniformly-sized chunks, facilitating better parallelization. Nonetheless, SLED incurs additional computational costs by appending a subset of tokens from the end of one chunk to the start of the next, serving as historical context. Building on SLED, LongLoRA (Chen et al., 2023a) applies the chunk pre-filling concept to the pre-training of long-text LLMs, using it to mitigate the exorbitant computational expenses during the pre-training phase. Built on the success of chunk prefilling, we investigate how well it works in practice.

More specifically, we divide $\boldsymbol{X} \in \mathbb{R}^{N \times D}$ into $M = N/S$ chunks, each of size $S$ (assuming $N$ is divisible by $S$), resulting in $\boldsymbol{X}_1, \cdots, \boldsymbol{X}_M$. Subsequently, let $T$ denote the index of the chunk currently being processed. For full attention, the output $\boldsymbol{O}_T$ (a submatrix of $\boldsymbol{O}$ corresponding to the outputs of the $T$-th chunk) is calculated as follows:

$$\boldsymbol{O}_T = \text{CausalAttention}(\boldsymbol{X}_T, \boldsymbol{X}_T, \boldsymbol{X}_T) + \sum_{t=1}^{T-1} \text{Attention}(\boldsymbol{X}_T, \boldsymbol{X}_t, \boldsymbol{X}_t). \quad (3)$$

We break the computation of causal attention over input $\boldsymbol{X}$ into multiple attentions over different input chunks. The Attention function in Eq. (3), similar to causal attention, is defined as

$$\text{Attention}(\boldsymbol{X}_T, \boldsymbol{X}_t, \boldsymbol{X}_t) = \text{softmax}(\boldsymbol{X}_T \boldsymbol{W}_q \boldsymbol{W}_k^\top \boldsymbol{X}_t^\top) \boldsymbol{X}_t \boldsymbol{W}_v, \quad (4)$$

but without the causal masking.

For block sparse attention, we are exploring whether the approximated $\boldsymbol{O}_T$ computed as

$$\boldsymbol{O}_T \approx \text{CausalAttention}(\boldsymbol{X}_T, \boldsymbol{X}_T, \boldsymbol{X}_T), \quad (5)$$

is sufficient. This approach clearly limits the receptive field of each token, but can be computed efficiently with standard PyTorch implementation. We will investigate the extent to which performance may be affected.

As depicted in Figure 4, block diagonal attention for pre-filling effectively preserves model performance without further fine-tuning, despite not computing full attention. However, as context length increases, its effectiveness diminishes, and the gap with full attention widens. This discrepancy is due to missing entries in our sparse attention matrices, but what should we focus on to compensate for the missing attention weights?

**Can We Recover Performance Loss Through Fine-tuning?** Since we are not computing the original full attention, the output will inevitably deviate from what would be produced using full attention. For example, due to the normalizing effect of the softmax function, the nonzero attention weights many be amplified compared to their original values. Additionally, some information from the off-diagonal block will be forfeited due to the imposed sparsity. We are exploring whether fine-tuning the model weights could counteract these alterations? In this scenario, there are two

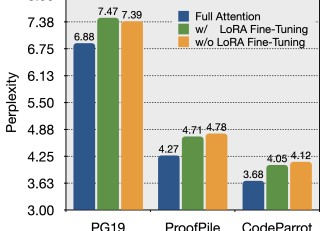

Figure 5: Fine-tuning with/without LoRA (Hu et al., 2022) on block sparse attention.

distinct phases: pre-filling and decoding, each employing different attention strategies for the tokens involved. This divergence from standard LLM pretraining introduces unique challenges. To avoid a surge in cost and complexity, we adhere to the standard LLM pretraining protocol. However, for each training input, we segment it to multiple chunks, and designating only the final chunk as generation tokens, while retaining all preceding chunks as pre-filling context tokens. The loss is calculated solely for the last chunk. Our observations indicate that fine-tuning marginally enhances performance, as illustrated in Figure 5. This suggests that while fine-tuning can offer some improvement, it may not fully compensate for the loss from not utilizing full attention.

## 4 Hierarchy-Aided Sparse Attention

In Sec. 3, we observe that fine-tuning with or without LoRA does not entirely mitigate the performance degradation resulting from the transition from full attention to block sparse attention. We recognize that the reduced computational complexity in Eq. (5), relative to Eq.( 3), is attributed to the attention mechanism across separate blocks, as illustrated:

$$\boldsymbol{R}_T = \sum_{t=1}^{T-1} \text{Attention}(\boldsymbol{X}_T, \boldsymbol{X}_t, \boldsymbol{X}_t). \tag{6}$$

Thus, the primary challenge lies in how to compensate for the loss of information without reverting to the full computation outlined in Eq. (3). In this section, we initially scrutinize a simplistic hard-coded hierarchical attention approximation to Eq. (6). Following this analysis, we introduce a specialized branch designed to reintegrate the omitted information.

### 4.1 Hierarchical Attention Approximation

In recent years, the advent of sparse attention mechanisms has spurred the development of innovative approaches to restore lost information via low-resolution or low-rank approximations. For instance, the H-Transformer-1D model (Zhu & Soricut, 2021) downsamples tokens based on their proximity to the sequence head, representing distant tokens at a lower resolution. Similarly, LongT5 (Guo et al., 2022) pools tokens beyond the immediate local window into global tokens using average pooling, then attends to these global tokens to recover omitted details.

Inspired by these methods, a straightforward strategy to approximate full attention is to maintain high fidelity for local interactions with higher-resolution representations, while using less precise approximations for distant interactions with lower-resolution representations. Specifically, for each segment $\boldsymbol{X}_t$, a summarized, low-resolution representation can be computed as follows:

$$\overline{\boldsymbol{x}}_t = \text{AvgPool}(\boldsymbol{X}_t). \tag{7}$$

Subsequently, the missing component could be approximated by

$$\boldsymbol{R}_T \approx \sum_{t=1}^{T-1} \text{Attention}(\boldsymbol{X}_T, \overline{\boldsymbol{x}}_t, \overline{\boldsymbol{x}}_t) \tag{8}$$

that captures the essence of integrating global information through a low-resolution approximation. We exploit a hierarchy of resolutions in attention approximation, and the final hierarchical approximation can be formulated as:

$$\boldsymbol{O}_T \approx \text{CausalAttention}(\boldsymbol{X}_T, \boldsymbol{X}_T, \boldsymbol{X}_T) + \sum_{t=1}^{T-1} \text{Attention}(\boldsymbol{X}_T, \overline{\boldsymbol{x}}_t, \overline{\boldsymbol{x}}_t), \tag{9}$$

which enhances the model's ability to manage long-term dependencies without excessive overheads.

Empirical evidence, as demonstrated in Table 1, suggests that incorporating global information through Eq. (9) does indeed partially restore model performance. However, there remains a performance gap compared to models utilizing full attention. Upon further analysis, it becomes evident that the potential cause of this discrepancy lies in the hard-coded operation where the chunk $\boldsymbol{X}_T$ is attended to $\overline{\boldsymbol{x}}_t$ with equal weight, contradicting the general consensus that strong attention scores are typically focused on specific tokens (Zhang et al., 2023b; Xiao et al., 2024).

Table 1: Results for Different Modifications Applied to LLaMA2-7B-32K. We compare three modifications: diagonal sparse attention (Diag. Sparse), hierarchical attention (Hie. Attn.), and hierarchical transformer (Hie. Tran.).

| Sequence Length | Diag. Sparse | Hie. Attn. | Hie. Tran. | LoRA | PG19 | ProofPile | CodeParrot |
|---|---|---|---|---|---|---|---|
| 8192 | | | | | 7.97 | 3.45 | 2.23 |
| | ✓ | | | | $9.21^{+1.24}$ | $4.05^{+0.60}$ | $2.60^{+0.37}$ |
| | ✓ | ✓ | | ✓ | $8.33^{+0.36}$ | $3.78^{+0.33}$ | $2.37^{+0.14}$ |
| | ✓ | | ✓ | ✓ | $7.22^{-0.75}$ | $3.33^{-0.12}$ | $2.11^{-0.12}$ |

## 4.2 HIERARCHICAL TRANSFORMER

To integrate hierarchical attention into a conventional attention mechanism, we developed a method adhering to the original computational framework. This introduces a specialized branch to process global information, forming the foundation of our hierarchical transformer.

Specifically, the specialized branch computes a global embedding representation $\tilde{X} = [\tilde{x}_1, \cdots, \tilde{x}_M] \in \mathbb{R}^{M \times D}$ that encapsulates the global interactions of the input sequence. Here, we assume that each embedding representation $\tilde{x}_t$ captures the interactions between the $t$-th chunk and other chunks. The computation of $\tilde{X}$ will be detailed shortly. For the $T$-th chunk, the component $R_T$ is recalculated, providing an alternative to Eq. (9), as follows:

$$R_T \approx \text{Attention}(X_T, \tilde{x}_T, \tilde{x}_T). \tag{10}$$

Then, the precise attention computation is defined as:

$$O_T \approx \text{CausalAttention}(X_T, X_T, X_T) + \text{Attention}(X_T, \tilde{x}_T, \tilde{x}_T). \tag{11}$$

This approach incorporates global information $\tilde{X}$ into each chunk while utilizing local attention, and facilitates information exchange among tokens within each chunk. Subsequently, $O_T$ is fed into the transformer's following modules in the standard manner.

To ensure the dedicated branch outputs a representation $\tilde{X}$ that accurately captures global interactions and is easily interpretable by the model, we proceed as follows. We take the embedding $X_{-1}$ from the previous layer $\text{Transformer}_{-1}$ and compute its pooled representation $\overline{X}_{-1}$ as:

$$\overline{X}_{-1} = \begin{bmatrix} \overline{x}_{-1,1} & \cdots & \overline{x}_{-1,M} \end{bmatrix}. \tag{12}$$

Then, we directly utilize the pre-trained layer of the same transformer to compute $\tilde{X}$:

$$\tilde{X} = \text{Transformer}_{-1}(\overline{X}_{-1}), \tag{13}$$

To prevent lookahead, which refers to the leakage of information within a chunk through low-resolution tokens as depicted in Figure 2, the causal attention mechanism has been subtly modified. This modification is achieved by introducing a shifted causal mask, which is created by removing the main diagonal from the original causal mask. We analyze that, in contrast to the hard-coded attention from Eq. (9), Eq. (10) also guides $X_T$ to attend to $\overline{x}_t$ with more adaptive weights, since $\tilde{x}_T \in \tilde{X}$ attends to $\overline{x}_t$ in a manner akin to regular causal attention.

Following these enhancements, we conducted LoRA fine-tuning on all projection matrices within the self-attention and MLP modules, ensuring the model's performance is optimized without extensive retraining. Empirical findings in Table 1 indicate that when the pretrained LLMs are augmented with our method, they exhibit exceptional performance and efficiency across various tasks.

The extensive preliminary research (Shen et al., 2018; Ye et al., 2019; Guo et al., 2019; Sukhbaatar et al., 2019) into hierarchical attention has confirmed its effectiveness. As the importance of long-context LLMs grows, the need for sub-quadratic hierarchical attention mechanisms has become more critical. However, a significant limitation of these hierarchical attentions is that they are not designed for LLMs, often requiring training from scratch, which is impractical in LLM environments. Our research builds upon the successes of previous methods while addressing their limitations, developing an innovative hierarchical attention mechanism that enables full performance recovery of pretrained LLMs through a few steps of LoRA fine-tuning.

Table 2: Perplexity of different pre-trained models on PG19, Proof-Pile, and CodeParrot datasets. Notably, LinPrefill-4K successfully reduces perplexity under 4K context length.

| Model | PG19↓ | | | ProofPile↓ | | | CodeParrot↓ | | |
|---|---|---|---|---|---|---|---|---|---|
| | 1K | 2K | 4K | 1K | 2K | 4K | 1K | 2K | 4K |
| LLaMA2-7B | **6.323** | 6.641 | 6.879 | **5.065** | 4.672 | 4.277 | **4.109** | 3.843 | 3.679 |
| Positive Control | 6.474 | 6.613 | 6.769 | 5.069 | 4.607 | 4.228 | 4.177 | 3.783 | 3.527 |
| LLaMA2-7B w/ HASA | 6.508 | **6.495** | **6.232** | 5.09 | **4.562** | **4.033** | 4.136 | **3.637** | **3.149** |

## 5 EXPERIMENTS

### 5.1 EVALUATIONS AND SETUPS

**Evaluations.** We conducted an extensive evaluation of HASA across several critical dimensions. 1) *Language Modeling*: Evaluated using datasets PG19 (Rae et al., 2019), Proof-Pile (Zhangir Azerbayev, 2022), and CodeParrot (L. et al., 2022), which collectively measure the model's ability to model language sequences. 2) *Few-shot Natural Language Understanding*: Our assessment utilized diverse benchmarks, including GLUE (Wang et al., 2018), SuperGLUE (Wang et al., 2019), OpenbookQA (Mihaylov et al., 2018), HellaSwag (Zellers et al., 2019), PiQA (Bisk et al., 2020), Winogrande (Sakaguchi et al., 2021), ARC-C, ARC-E (Clark et al., 2018), and MathQA (Amini et al., 2019). 3) *Long-Context Downstream Tasks*: We employed benchmarks including Needle in a Haystack (gkamradt, 2023), and LongBench (Bai et al., 2024) to assess the model's proficiency in understanding and generating content within extended-context scenarios. This multidimensional evaluation strategy ensures a thorough examination of our method's strengths and weaknesses across a range of complex and challenging environments.

**Setups.** We adapted two base models: LLaMA2-7B (Touvron et al., 2023b) with a chunk size of $S = 128$ and LLaMA2-7B-32K (Together.AI, 2023a) with a chunk size of $S = 1024$. We applied LoRA (Hu et al., 2022) to all projection matrices within the self-attention and MLP modules, using hyperparameters $r = 16$, $\alpha = 32$. 1) For LLaMA2-7B with HASA, we utilized 100,000 samples from the SlimPajama dataset (Soboleva et al., 2023) for training. Concurrently, we fine-tuned the original LLaMA2-7B model on the same dataset with identical LoRA parameters to serve as a positive control. Both LLaMA2-7B with HASA and the positive control models were trained with a learning rate of 1e-4, a batch size of 8, the Adam optimizer without weight decay, and a cosine scheduler. 2) For LLaMA2-7B-32K with HASA, to facilitate comparison with other chat assistants, we implemented a two-stage training recipe. After an initial training phase with 100,000 samples from the SlimPajama dataset (Soboleva et al., 2023), we conducted a second-stage instruction fine-tuning. This stage incorporated a mixed dataset, comprising LongAlpaca (Chen et al., 2023b) (55.5%), Single-Detail QA (Zhang et al., 2024b)[1] (30%), BookSum (Kryściński et al., 2021) (12%), and Needle (2.5%). These datasets were formatted into conversational formats for fine-tuning using the Vicuna chat template (Zheng et al., 2023). The optimizer settings for LLaMA2-7B-32K with HASA mirrored those for LLaMA2-7B, except for a reduced learning rate of 2e-5.

### 5.2 PERFORMANCE COMPARISON

**Language Modeling.** We evaluated the language modeling capabilities of LLaMA2-7B with HASA on three datasets: PG19, ProofPile, and CodeParrot. These datasets cover three distinct domains—books, mathematics, and code—offering a comprehensive assessment. For PG19, we used 100 samples from the test set, while for ProofPile, we utilized 79 samples randomly selected according to the method described by Zhang et al. (2024a). For the CodeParrot dataset, we concatenated code from the same repository and then sampled 100 examples for evaluation, following the RPT (Rubin & Berant, 2023) and LLM-Embedder (Zhang et al., 2023a) methodologies. The results are shown in Table 2. It is evident that LLaMA2-7B with HASA significantly outperforms the LLaMA2-7B on input text lengths of 2K and 4K, with a large margin. Remarkablly, it even surpasses the positive control, which is a full-attention baseline.

---

[1]Single-Detail QA involves using GPT-4 to generate question-answer pairs from long contexts, *e.g.*, a book or a lengthy paragraph, for dataset construction.

**Few-Shot Natural Language Understanding.** We evaluated the model using the GLUE, SuperGLUE, OpenBookQA, HellaSwag, PiQA, Winogrande, ARC-C, ARC-E, and MathQA benchmarks, all under a 5-shot configuration. As shown in Table 3, LLaMA2-7B with HASA generally maintains the natural language understanding capabilities compared to the baselines.

Table 3: Natural language understanding results. LLaMA2-7B w/ HASA maintains its performance using a chunk size of $S = 128$. The symbol † denotes the mismatched version of MNLI.

| BenchMark | | Metric | LLaMA2-7B | Positive Control | LLaMA2-7B w/ HASA |
|---|---|---|---|---|---|
| **GLUE @5-shot** | CoLA | mcc | 0.243 | 0.168 | 0.17 |
| | MNLI | acc | 0.376 | 0.356 | 0.478 |
| | †MNLI | acc | 0.424 | 0.376 | 0.502 |
| | MRPC | acc | 0.681 | 0.683 | 0.674 |
| | RTE | acc | 0.657 | 0.638 | 0.599 |
| | QNLI | acc | 0.483 | 0.472 | 0.483 |
| | QQP | f1 | 0.582 | 0.458 | 0.466 |
| | **AVERAGE** | - | **0.492** | **0.450** | **0.481** |
| **SuperGLUE @5-shot** | BoolQ | acc | 0.769 | 0.746 | 0.746 |
| | CB | acc | 0.571 | 0.482 | 0.563 |
| | COPA | acc | 0.9 | 0.88 | 0.88 |
| | MultiRC | acc | 0.435 | 0.433 | 0.445 |
| | ReCoRD | f1 | 0.916 | 0.91 | 0.905 |
| | WiC | acc | 0.562 | 0.517 | 0.51 |
| | WSC | acc | 0.365 | 0.365 | 0.394 |
| | **AVERAGE** | - | **0.645** | **0.619** | **0.634** |
| **@5-shot** | OpenbookQA | acc | 0.342 | 0.34 | 0.35 |
| | HellaSwag | acc | 0.604 | 0.592 | 0.581 |
| | PiQA | acc | 0.775 | 0.779 | 0.768 |
| | Winogrande | acc | 0.728 | 0.727 | 0.722 |
| | ARC-C | acc | 0.477 | 0.462 | 0.469 |
| | ARC-E | acc | 0.793 | 0.789 | 0.788 |
| | MathQA | acc | 0.26 | 0.258 | 0.267 |
| | **AVERAGE** | - | **0.568** | **0.563** | **0.564** |

**LongBench.** To evaluate generalization ability on downstream tasks, we utilized five sub-tasks from the LongBench (Bai et al., 2024): SingleDoc QA, MultiDoc QA, Summarization, Few-shot, and Code Completion. For this benchmark, the model first needs to understand a long context and then generate a response based on the given instruction. The evaluation is conducted by directly comparing the overlap between the model's response and the ground truth. As such, this task provides a comprehensive assessment of the model's understanding and generation capabilities. For LLaMA2-7B with HASA, we selected LLaMA2-7B (Touvron et al., 2023b) and the positive control model fine-tuned using the same recipe as baselines. For LLaMA2-7B-32K with HASA, we chose several popular models based on LLaMA2-7B or LLaMA2-7B-chat as baselines, namely LLaMA2-7B-32K (Together.AI, 2023a), LLaMA2-7B-32K-Instruct (Together.AI, 2023b), LongChat-7B-v1.5-32K (Li et al., 2023), LongAlpaca-7B-16K (Chen et al., 2023a), Vicuna-v1.5-7B-16K (Zheng et al., 2024), YaRN-7B-128K (Peng et al., 2024), and Activation Beacon (Zhang et al., 2024a). In addition, we used GPT-3.5-Turbo-16K (Achiam et al., 2023) as a reference model.

For all models with a context window smaller than 16K, we applied left-side truncation whenever the token sequence exceeded the maximum allowable length. Table 4 presents the average scores across five sub-tasks and the end-to-end pre-filling latency on the initial dataset for the first sub-task.

As shown in the table, both LLaMA2-7B with HASA and LLaMA2-7B-32K with HASA achieve performance comparable to or better than other baseline models. Notably, LLaMA2-7B-32K with HASA excels in both SingleDoc QA and MultiDoc QA, significantly outperforming other LLaMA2-7B based chat assistants. These downstream experimental results indicate that using sparse attention during the pre-filling stage does not compromise the model's ability to understand context or generate responses. Additionally, compared to other 32K models, LLaMA2-7B-32K with HASA achieves

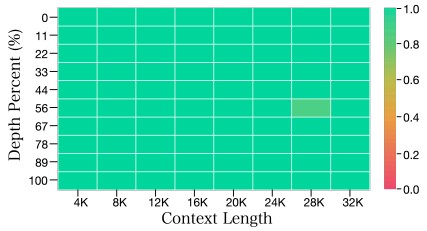

Figure 6: Results of Needle In A Haystack test.

Table 4: The performance of various models on the five sub-tasks of LongBench. † denotes the results as reported by Zhang et al. (2024a).

| Model | SQA | MQA | SUM | FEW | CODE | End-to-End Latency |
|---|---|---|---|---|---|---|
| **Context Length = 4K** | | | | | | |
| LLaMA2-7B | 14.54 | 8.73 | 7.7 | 48.94 | 58.75 | 4m20s $^{1.15\times}$ |
| Positive Control | 13.56 | 7.94 | 19.56 | 48.54 | 56.62 | 4m21s $^{1.15\times}$ |
| LLaMA2-7B w/ HASA | 14.34 | 9.98 | 19.24 | 48.3 | 57.22 | 3m46s $^{1.00\times}$ |
| **Context Length ⩾ 16K** | | | | | | |
| LLaMA2-7B-32K | 2.88 | 8.14 | 4.95 | 58.06 | 61.00 | 33m12s $^{1.56\times}$ |
| LLaMA2-7B-32K-Inst | 12.48 | 14.12 | 22.44 | 58.19 | 54.51 | 33m57s $^{1.60\times}$ |
| LongChat-7B-32K | 31.63 | 23.35 | 21.77 | 49.32 | 54.92 | 34m21s $^{1.62\times}$ |
| LongAlpaca-7B-16K | 26.66 | 28.01 | 24.57 | 52.99 | 52.48 | 18m51s $^{0.88\times}$ |
| GPT-3.5-Turbo-16K | 45.10 | 36.23 | 23.90 | 52.99 | 54.15 | - |
| Vicuna-v1.5-7B-16K | 31.75 | 18.80 | 23.25 | 57.58 | 47.25 | 18m46s $^{0.88\times}$ |
| †YaRN-7B-128K | 24.03 | 24.11 | 19.82 | 56.83 | 62.73 | 33m51s $^{1.59\times}$ |
| †Activation Beacon | 28.27 | 28.44 | 25.15 | 61.00 | 57.75 | 33m48s $^{1.59\times}$ |
| LLaMA2-7B-32K w/ HASA | 39.74 | 30.17 | 22.15 | 57.18 | 54.83 | 21m12s $^{1.00\times}$ |

approximately 1.5x faster pre-filling on the initial dataset, which has an average length of 18,409 tokens. More detailed scores can be found in Appendix B.

**Needle In A Haystack.** This test works by embedding specific, targeted information (the "needle") within a large, more compleix body of text (the "haystack"). The goal is to assess an LLM's ability to identify and utilize this specific piece of information amidst a vast amount of data. As illustrated in Figure 6, our method effectively retains the ability to process information at different positions across various context windows, ranging from 1K to 32K tokens.

## 5.3 EFFICIENCY ANALYSIS

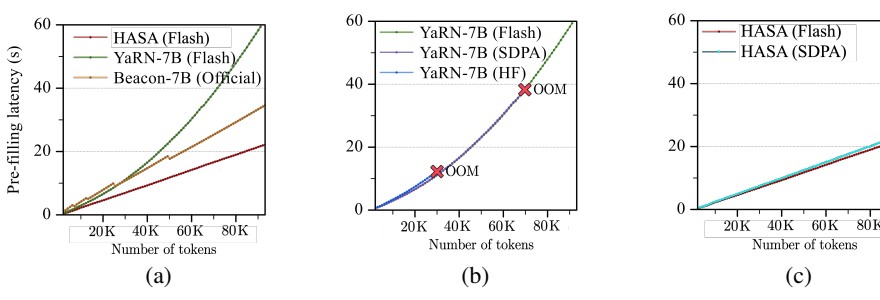

Figure 7: (a) Pre-filling latency comparison between HASA and others. (b) and (c) Impact of different attention implementations on latency.

The comprehensive benchmark results demonstrate the strong performance of HASA. In this subsection, we focus on its efficiency, specifically assessing whether it achieves our goal of significantly reducing pre-filling latency and addressing the quadratic scalability of TTFT.

**Theoretically, HASA Should Exhibit Nearly Linear Pre-filling Latency.** The low-resolution branch handles fewer tokens, reducing its quadratic term by a factor of $S^2$ compared to the full attention baseline. Consequently, the main latency arises from the high-resolution branch, where latency grows linearly with sequence length.

**HASA Demonstrates Near-Linear Pre-filling Latency in Practice.** We conducted experiments with an 8×RTX3090 machine to evaluate the pre-filling latency scaling behavior. For comparison, we included YaRN-7B (Peng et al., 2024), a full attention model without any acceleration techniques, and Activation Beacon (Zhang et al., 2024a), an efficient model based on LLaMA2-7B (Touvron et al., 2023a). To ensure a fair comparison, we evaluated YaRN-7B and LLaMA2-7B with HASA using different attention implementations, including Flash Attention 2 (Dao, 2024) (Flash), PyTorch SDPA (Guessous, 2024) (SDPA), and the default attention

implementation from HuggingFace Transformers (Wolf et al., 2020) (HF). As shown in Figure 7(a), compared to the full attention baseline, our method reduces pre-filling latency by $1.55\times$ for a sequence length of 32K and by $2.7\times$ for a sequence length of 96K. Since the latency of full attention grows quadratically, the speedup becomes more pronounced as the sequence length increases. On the other hand, even when compared to Activation Beacon, which exhibits near-linear latency growth, our method still demonstrates a significant reduction.

## 5.4 ABLATIONS

**Ablation on Chunk Size.** We selected three different chunk sizes—64, 256, and 2048—to investigate whether HASA's performance differs under smaller or larger local windows. We applied the same training recipe and evaluated the resulting models on LongBench (Bai et al., 2024). The results are shown in Table 5. As shown, when the local information is too limited, HASA's performance

Table 5: Ablation on chunk size.

| Chunk Size | SQA | MQA | SUM | FEW | CODE |
|---|---|---|---|---|---|
| $S = 64$ | 28.58 | 22.15 | 18.56 | 46.14 | 49.75 |
| $S = 256$ | 36.96 | 26.47 | 21.65 | 57.41 | 54.58 |
| $S = 2048$ | **40.23** | 28.82 | **24.91** | **58.73** | 54.83 |
| $S = 1024$ | 39.74 | **30.17** | 22.15 | 57.18 | **55.65** |

drops significantly across all sub-tasks. Additionally, we observe that 1024 serves as a sweet spot—chunk sizes larger than this threshold do not provide noticeable performance gains and, in fact, result in performance degradation on tasks like MultiDoc QA and Code. This suggests that most downstream tasks contain only sparse long-term dependencies, and as long as the local dependencies are accurately captured, performance can be maintained.

## 6 LIMITATION

First, while HASA can accelerate the pre-filling phase, this advantage becomes more evident with lengthy sequences. Second, although HASA offers efficient training, it is not a training-free method and still requires some computational resources, limiting its applicability compared to fully training-free approaches.

## 7 CONCLUSION

In this work, we revisited efficient pre-filling strategies and demonstrated that applying diagonal block sparse attention during the pre-filling phase effectively reduces computational costs by over 90%, without significant degradation in language modeling performance. To further close the performance gap between this naive method and full-attention pre-filling, we introduced Hierarchy-Aided Sparse Attention (HASA), which leverages a specialized branch to recover global information discarded by sparse attention. By processing global embeddings across chunks and calibrating attention scores, HASA stabilizes the sparse attention computation, allowing the model to handle long sequences while maintaining robust language modeling performance.

Our implementation of HASA on LLaMA2-7B and LLaMA2-7B-32K, fine-tuned with LoRA, demonstrated improved pre-filling efficiency and significantly reduced time to first token (TTFT) across a variety of benchmarks, including language modeling, few-shot NLU tasks, Needle in a Haystack, and LongBench. These results confirm that HASA not only accelerates pre-filling but also sustains, and in some cases, enhances model performance.

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

## A  IMPLEMENTATION DETAILS

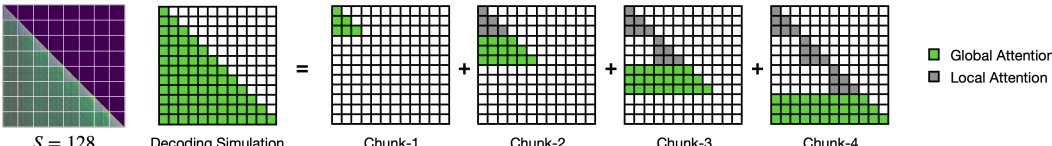

$S = 128$  Decoding Simulation  = Chunk-1 + Chunk-2 + Chunk-3 + Chunk-4

■ Global Attention
■ Local Attention

Figure 8: The decoding simulation can construct an attention matrix in which each token is endowed with global attention.

**Decoding Simulation.** Since our focus is solely on the decoding stage—specifically on the output and internal properties of tokens with global attention—applying language modeling loss to tokens from the pre-filling stage is unnecessary, as their key-value representations serve only as contextual embeddings, not for generating output. To improve data utilization, we aim to enable every token to function as a decoding-stage token for language modeling. Achieving this within a single forward pass is insufficient, as a token cannot simultaneously serve as both a pre-filling and decoding-stage token in the same pass. Therefore, we propose a two-pass approach. In the first forward pass, all tokens are treated as pre-filling-stage tokens, generating an approximate key-value (KV) cache at each layer. In the second pass, every token acts as a decoding-stage token, attending to the KV cache generated from the first pass. This decoding simulation approach, illustrated in Figure 8, is employed in both LLaMA2-7B with HASA and LLaMA2-7B-32K with HASA architectures.

**Position Embedding within a Chunk.** For RoPE (Su et al., 2021) within a chunk, there are two encoding strategies. The first involves encoding based on the token's position within the original sequence, while the second encodes the token relative to its position within the chunk itself. In post-training settings, both strategies deliver comparable performance. As a result, we adopted the second approach for both LLaMA2-7B with HASA and LLaMA2-7B-32K with HASA, as it offers easier implementation. However, in training-free scenarios, the choice of position encoding has a more significant impact. Specifically, using the first method can lead to model instability, whereas the second method maintains robust performance. This behavior is analyzed in detail by Xiao et al. (2024), who propose adding a sink token to each chunk as a solution. This modification allows the first encoding method to regain its stability and match the performance of the second method in training-free setups, achieving near lossless results.

## B  ADDITIONAL EXPERIMENTAL RESULTS

**Detailed Results on LongBench.** In Table 6, we present the detailed scores for each dataset across five sub-tasks from LongBench, along with the average number of tokens and other relevant information. For the baseline models, we selected LLaMA2-7B-32K (Together.AI, 2023a) (LM), LongChat-7B-32K (Li et al., 2023) (LC), LongAlpaca-7B-16K (Chen et al., 2023a) (LA), YaRN-7B-128K (Peng et al., 2024) (YN), and Activation Beacon (Zhang et al., 2024a) (AB). Since LongAlpaca-7B-16K has a context window smaller than 32K, left-side truncation was applied.

**Attention Visualization.** We aim for the approximated key-value representations generated during the pre-filling phase by our method to receive an attention allocation in the decoding stage that is comparable to that of the accurate key-value representations produced by the full attention baseline.

Table 6: Detailed results of various long-context LLMs on LongBench (Bai et al., 2024).

| Benchmark | | Avg Length | Lang | Metric | LM | LC | LA | YN | AB | LLaMA2-7B w/ HASA |
|---|---|---|---|---|---|---|---|---|---|---|
| **SingleDoc QA** | NarrativeQA | 18,409 | en | F1 | xx | 19.15 | 19.10 | 6.93 | 19.10 | **19.41** |
| | Qasper | 3,619 | en | F1 | xx | 29.41 | 26.73 | 12.36 | 19.33 | **39.14** |
| | MultiField-en | 4,559 | en | F1 | xx | 42.90 | 34.90 | 22.55 | 24.43 | **54.11** |
| | MultiField-zh | 6,701 | zh | F1 | xx | 35.06 | 25.91 | 16.93 | 20.91 | **43.94** |
| **MultiDoc QA** | HotpotQA | 9,151 | en | F1 | xx | 33.05 | **42.42** | 9.23 | 12.06 | 39.99 |
| | 2WikiMQA | 4,887 | en | F1 | xx | 24.14 | **35.28** | 9.56 | 13.46 | 29.96 |
| | MuSiQue | 11,214 | en | F1 | xx | 14.75 | **19.83** | 5.86 | 6.07 | 18.80 |
| | DuReader | 15,768 | zh | Rouge-L | xx | 21.47 | 14.50 | 17.83 | 17.94 | **23.30** |
| **Summarization** | GovReport | 8,734 | en | Rouge-L | xx | 30.83 | **31.38** | 24.49 | 27.51 | 28.09 |
| | QMSum | 10,614 | en | Rouge-L | xx | 22.93 | **23.98** | 18.84 | 22.56 | 23.06 |
| | MultiNews | 2,113 | en | Rouge-L | xx | 26.65 | **27.13** | 18.81 | 26.17 | 26.31 |
| | VCSUM | 15,380 | zh | Rouge-L | xx | 6.68 | **15.79** | 8.16 | 13.56 | 9.29 |
| **Few-shot** | TREC | 5,177 | en | Acc (CLS) | xx | 66.50 | 60.50 | 65.50 | 47.50 | **69.00** |
| | TriviaQA | 8,209 | en | F1 | xx | 83.99 | **85.27** | 84.01 | 77.25 | 81.43 |
| | SAMSum | 6,258 | en | Rouge-L | xx | 22.32 | 39.47 | 25.24 | 39.11 | **45.65** |
| | LSHT | 22,337 | zh | Acc (CLS) | xx | 24.50 | 26.75 | **30.5** | 10.50 | 29.00 |
| **Code Completion** | LCC | 1,235 | Python/C#/Java | Edit Sim | xx | 52.98 | 54.86 | **61.19** | 56.51 | 52.96 |
| | RepoBench-P | 4,206 | Python/Java | Edit Sim | xx | **56.86** | 50.10 | 55.48 | 49.72 | 49.40 |

To evaluate this, we visualized the attention maps produced by HASA during the decoding stage, along with those generated by the full attention baseline. Additionally, we also visualized their residuals. We utilized a decoding simulation chunk size of 128, using the first sample from the PG19 test set as the input sequence. The results are presented in Figure 9. Notably, for each layer, the attention maps generated by our method exhibit a strong similarity to those of the baseline, both on a global scale and at a more localized level. A comparison of the residuals reveals no significant systematic errors; rather, the differences appear to be akin to random noise. Specifically, in the attention map of layer 31, it is evident that our method effectively attends to several significant tokens, indicating its capability to capture long-term dependencies.

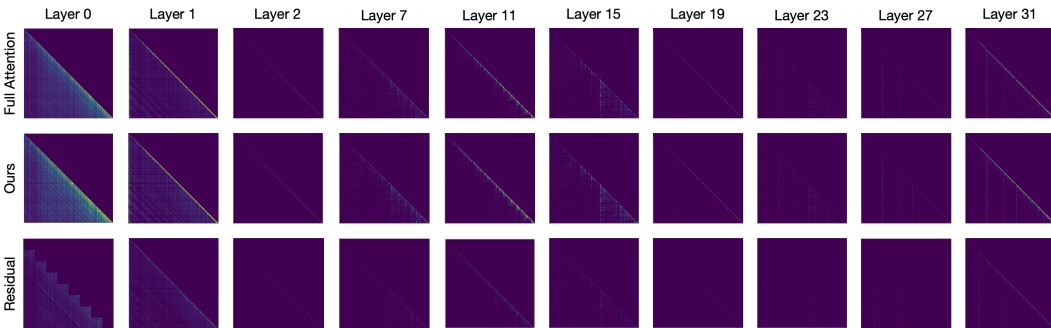

Figure 9: Attention map visualization for a subset of layers, with head 0 displayed for each layer.

