# OpenReview forum: "Hierarchy-Aided Sparse Attention For Fast LLMs Prefilling Inference"
_ICLR.cc/2025/Conference — Submitted to ICLR 2025_

### Official Review · Reviewer_Qi4G · 2024-11-03

**Soundness:** 3
**Presentation:** 4
**Contribution:** 3
**Rating:** 5
**Confidence:** 4

**Summary:**

This paper introduces Hierarchy-Aided Sparse Attention (HASA), a simple and effective approach to enhance the efficiency of the pre-filling phase in large language models (LLMs) when processing long-context inputs. By employing diagonal block sparse attention with specifically designed global attention tokens, they significantly reduce attention-related floating-point operations (FLOPs) without largely compromising language modeling performance.

**Strengths:**

1. This paper is well-written, with a clear analysis and step-by-step presentation of the improvement from the original diagonal block sparse attention to the Hierarchy-Aided Sparse Attention. Hierarchy-Aided Sparse Attention is technically sound, and the experiments demonstrate its efficiency in the pre-filling phase with a long-context input.
2. The experiments are thoroughly conducted on LM, few-shot tasks, and long-context tasks, showing the effectiveness of Hierarchy-Aided Sparse Attention.

**Weaknesses:**

1. Introducing another transformer branch to distill the global information seems computationally heavy. It's better to include the original diagonal block sparse attention and HASA w.o. another transformer branch as baselines for ablating the influence of each proposed component for efficiency.
2. A strong baseline is sliding window attention plus some global attention at the end. In this way, information flows between overlapped blocks, compared to the independent block sparse attention. I would like to see the efficiency and performance compared to HASA.
3. Why does HASA not include the sink attention? As many papers suggest, the LLMs tend to contribute most of the attention score to the first token (i.e., the sink token). It seems that more efforts will be needed (e.g., training cost and network design) to convert full attention to sparse attention when dropping the sink token.

**Questions:**

Please address the concerns and questions in the Weaknesses part.

---

### Official Review · Reviewer_nVWr · 2024-11-04

**Soundness:** 2
**Presentation:** 2
**Contribution:** 2
**Rating:** 5
**Confidence:** 4

**Summary:**

This paper introduces HASA, hierarchy-aided sparse attention, to reduce the quadratic computation complexity. HASA incorporates a specialized transformer branch that combines sparse attention with hierarchical embeddings. Specifically, HASA first separate long texts into a series of chunks and compute local attention for each chunk during prefilling. To setup the connection between different chunks, the tokens inside each chunk also attend to a special embedding, which is an average pooling of token embeddings inside the chunk and attends to other previous special embeddings via the specialized branch with shift mask attention. Experiments conducted on various benchmarks demonstrate that HASA not only maintains performance but also outperforms baseline methods in certain scenarios.

**Strengths:**

- The key idea of leveraging a special embedding to capture dependency among different chunks is interesting in the context of efficient pre-filling.
- HASA achieves lower end-to-end latency compared to baseline methods.
- Table 4 shows that HASA achieves superior performance on several tasks from LongBench such as SQA and MQA.

**Weaknesses:**

1. HASA introduces a specialized branch to model inter-chunk attention, which, while ensuring efficiency, increases GPU memory consumption.
2. The performance comparison experiments with prior methods are not fully convincing. For example, YaRN-7B was trained on the PG19 dataset, whereas HASA was trained on the SlimPajama dataset in stage 1, followed by stage 2 instruction tuning on a mixed dataset. It is recommended to reimplement YaRN using the datasets utilized in this paper for a fairer comparison.
3. Compared to training-free methods for efficient long-context inference, such as Streaming LLM and MInference, HASA's adaptability to various LLMs and scenarios may be limited.

**Questions:**

Please see Weakness.

---

### Official Review · Reviewer_FJML · 2024-11-04

**Soundness:** 3
**Presentation:** 2
**Contribution:** 3
**Rating:** 5
**Confidence:** 3

**Summary:**

This paper proposes using block sparse attention to reduce prefilling time and an additional branch of global attention to compensate for the loss of global information. For global attention, they pool each block to get the representative token for each block and pass it to a new branch of the transformer.  This paper further parameter efficient fine-tuning LLMs with this method. This method improves over full attention on long context benchmarks and reduces latency.

**Strengths:**

1. This paper proposes a novel way to bridge sparse attention methods and hierarchical attention.

2. The prefilling phase is time-consuming, and this paper reduces time while maintaining most of the performance.

**Weaknesses:**

1. The novelty of this paper is limited. This paper is a mix of existing methods.

2. This method's speedup is not significant, and the additional branch takes twice the time to pass transformed layers.

**Questions:**

1. In Figure 2, it seems you are trying to express information with color; please add descriptions of each color.

---

### Meta-Review · Area_Chair_Ak8x · 2024-12-23

**Metareview:**

### Summary
This paper proposes a method (Hierarchy-Aided Sparse Attention) to accelerate the pre-filling phase of large language models (LLMs) with long-context inputs. HASA reduces the quadratic complexity of full attention by leveraging block sparse attention and introduces a specialized transformer branch to handle global information across chunks. The method achieves significant speedups in the pre-filling phase while maintaining strong performance on long-context benchmarks.

### Strengths
The proposed method combines sparse and hierarchical attention, which provides an efficient mechanism for pre-filling acceleration. The experiments demonstrate robust performance on various long-context tasks, and the paper is clearly written with thorough analysis and insights.

### Weaknesses:
As noted by reviewers, HASA has limited novelty compared to existing techniques, and the additional transformer branch increases computational overhead and memory usage. The comparison with baseline methods lacks fairness due to differences in training datasets. Some key ablations and baseline comparisons, such as diagonal sparse attention without the additional transformer branch, are missing. Furthermore, concerns regarding the adaptability of HASA across broader scenarios were not adequately addressed.

**Additional Comments On Reviewer Discussion:**

During the rebuttal period, the reviewers raised the following key points:

1. Baselines: The reviewers noted that comparisons with YaRN-7B were unfair due to differences in training datasets. There are also other missing important baselines.

2. Ablation Studies: The reviewers requested additional ablations to isolate the impact of each component of HASA, such as diagonal sparse attention without the transformer branch.

The authors did not provide a rebuttal or any clarification on these issues.

---

### Decision · Program_Chairs · 2025-01-22

Reject